# Effect of *Poria cocos* Terpenes: Verifying Modes of Action Using Molecular Docking, Drug-Induced Transcriptomes, and Diffusion Network Analyses

**DOI:** 10.3390/ijms25094636

**Published:** 2024-04-24

**Authors:** Musun Park, Jin-Mu Yi, No Soo Kim, Seo-Young Lee, Haeseung Lee

**Affiliations:** 1Korean Medicine (KM) Data Division, Korea Institute of Oriental Medicine, Daejeon 34054, Republic of Korea; 2KM Convergence Research Division, Korea Institute of Oriental Medicine, Daejeon 34054, Republic of Korea; jmyi@kiom.re.kr (J.-M.Y.); nosookim@kiom.re.kr (N.S.K.); 3KM Science Research Division, Korea Institute of Oriental Medicine, Daejeon 34054, Republic of Korea; 09seoyoung03@kiom.re.kr; 4College of Pharmacy, Pusan National University, Busan 46241, Republic of Korea; haeseung@pusan.ac.kr

**Keywords:** *Poria cocos*, terpenes, molecular docking, transcriptome, diffusion network, Alzheimer’s disease, antioxidant

## Abstract

We characterized the therapeutic biological modes of action of several terpenes in *Poria cocos* F.A Wolf (PC) and proposed a broad therapeutic mode of action for PC. Molecular docking and drug-induced transcriptome analysis were performed to confirm the pharmacological mechanism of PC terpene, and a new analysis method, namely diffusion network analysis, was proposed to verify the mechanism of action against Alzheimer’s disease. We confirmed that the compound that exists only in PC has a unique mechanism through statistical-based docking analysis. Also, docking and transcriptomic analysis results could reflect results in clinical practice when used complementarily. The detailed pharmacological mechanism of PC was confirmed by constructing and analyzing the Alzheimer’s disease diffusion network, and the antioxidant activity based on microglial cells was verified. In this study, we used two bioinformatics approaches to reveal PC’s broad mode of action while also using diffusion networks to identify its detailed pharmacological mechanisms of action. The results of this study provide evidence that future pharmacological mechanism analysis should simultaneously consider complementary docking and transcriptomics and suggest diffusion network analysis, a new method to derive pharmacological mechanisms based on natural complex compounds.

## 1. Introduction

*Poria cocos* F.A. Wolf (PC), or Bokryeong, is a type of fungus that has long been used to treat a variety of diseases [1]. As terpenes present in PC exert multiple pharmacological effects [2,3], PC is highly valued as a therapeutic agent. It has been used to treat edema caused by water retention in the body [4]. Because PC contains active compounds, it is used in traditional herbal remedies, such as Oryeong-san which exhibits a diuretic effect [5], Samryeongbaekchul-san which exerts an antidiarrheal effect [6], Yukgunja-tang which enhances digestive function [7], and Gwibi-tang which has a tranquilizing effect [8].

PC possesses anti-tumor and anti-inflammatory properties and affects blood pressure, diabetes, immune regulation, and Alzheimer’s disease (AD) [9]. Therefore, determining the modes of action (MOAs) of PC is essential to understanding the efficacy of the fungus in treating various diseases.

Generally, herbal medicines use plants, animals, and minerals but fungi are rarely used. Mushrooms contain high amounts of lanostane and seco-lanostane terpenes; therefore, they may have pharmacological mechanisms different from those of other herbal medicines [3].

Drugs with similar scaffolds have similar mechanisms of action, indicating that the unique compounds in PC may have unique mechanisms [10,11,12]. Therefore, studying the PC-specific MOA goes beyond simply studying the MOA of herbal medicines and exploring new MOAs that are not generally associated with herbal medicines.

As PC contains unique compounds, the results of MOA prediction differ for other herbal medicines and those for PC compounds. However, few studies have predicted the MOA of PC based on in silico experiments using PC compounds. Although reviews on PC compounds and their MOA have been published [13,14,15], comprehensive MOA studies have not yet been conducted to elucidate the effects of PC. Therefore, studying the various biological MOAs of the unique compounds of PC on ailments, such as diuresis, diarrhea, digestion, and tranquilization is necessary.

Several methods can be used to predict the broad MOA of herbal medicines, such as molecular docking analysis (MDA) and drug-induced transcriptome analysis (DTA). MDA predicts whether a specific compound interacts with a specific protein by calculating the potential energy function and force field based on the molecular energy calculated from the structures of the compound and protein [16]. DTA confirms drug response by measuring the relative expression of mRNA in cells or specific tissues [17]. The use of these methods can elucidate the various MOAs of PC.

Although MDA and DTA are excellent methods, they have several drawbacks. MDA can predict direct drug-target interactions but not downstream drug actions. DTA can confirm the transcription resulting from drug treatment but not the upstream drug action. MDA and DTA have clear disadvantages but because they are complementary, it is expected that the complex MOA of herbal medicines will be revealed if the two methods are used together.

As the MOAs of the terpenes of PC are underexplored, we aimed to study the various biological MOAs of the terpenes contained in PC. The upstream terpene mechanism was predicted by drug–protein interactions through MDA. DTA using brain and colon cell lines was performed to confirm the downstream mechanism of PC and identify the therapeutic effects of PC on neuropsychiatric and digestive diseases. Finally, an analytical method based on a diffusion network (DN) that connects the upstream and downstream mechanisms of PC was proposed [18], and the detailed pharmacological MOA of PC in AD pathology was confirmed.

## 2. Results

### 2.1. Data on Terpenes Collected from PC

The data on terpenes in PC were collected from several research papers and herb-compound databases, and the compounds were divided into six groups. Each group contained 123 lanostane-type triterpenes, 51 seco-lanostane-type triterpenes, 6 tricyclic diterpenes, 5 tricyclic diterpenes, 20 sterols, and 15 other compounds (Appendix A). Among the collected terpenes, 125 druggable compounds with 3D molecular structures were searched in the PubChem Database.

### 2.2. Molecular Docking Using Terpenes

#### 2.2.1. Over-Representation Analysis (ORA) Using Unique Terpenes of PC

Twenty proteins with docking probabilities >50% in the lanostane and seco-lanostane groups and <50% in the non-lanostane groups were selected (Table 1). ORA using these 20 proteins predicted them to be effective in treating neurological and psychiatric diseases (Figure 1). Brain-related pathways, such as the Rap1 signaling pathway, nicotine addiction, neuroactive ligand–receptor interaction, and cocaine addiction, were identified using the Kyoto Encyclopedia of Genes and Genomes (KEGG) (Figure 1A). Synapse-related actions and interleukin actions were confirmed using Gene Ontology (GO) biological processes (Figure 1B); AD and hypertension were confirmed using the Online Mendelian Inheritance in Man (OMIM) database (Figure 1C). Visualization of docking analysis (Appendix A) revealed the interaction of three compounds at the same position in ITGAL but the number of atoms interacting with the amino acid was different due to the difference in the size of the scaffold of compounds. Therefore, 7-Oxodehydroabietic acid, a non-lanostane compound, may preferentially act on proteins other than ITGAL despite functioning at the same location.

#### 2.2.2. ORA Using Main Dockable Proteomes

As a result of the ORA using the main dockable proteomes, 54 pathways were effective (Table 2 and Appendix A). These 54 pathways showed various biological MOA possessed by PC. Biological terms such as the calcium signaling pathway, hypertrophic cardiomyopathy, and aldosterone synthesis and secretion were related to hypertension. Cortisol synthesis and secretion were associated with type II diabetes mellitus. Serotonergic synapses, gamma-aminobutyric acid (GABA), ergic synapses, neurodegenerative pathways, and AD were biological terms related to neurological and psychiatric diseases.

#### 2.2.3. GSEA Using Drug-Induced Transcriptomes

GSEA based on the PC-induced transcriptomes of the SW1783 cell line revealed that many pathways were functional (Figure 2). We found that PC was effective against neuropsychiatric diseases, such as AD, Parkinson’s disease, and Huntington’s disease at low and intermediate concentrations. It acted as effectively as MDA in hypertrophic cardiomyopathy but did not affect the calcium signaling, aldosterone-related, and diabetes-related pathways. In the hypertrophic cardiomyopathy pathway, it functioned as effectively as MDA. However, it did not affect the calcium signaling, aldosterone, or diabetes-related pathways, showing different results from those of MDA. In addition, it was confirmed that PC was effective against bacterial infections by examining pathways such as *Vibrio cholerae* infection, pathogenic *Escherichia coli* infection, and viral myocarditis. GSEA of HT29 cells revealed many biological terms related to metabolism (Appendix A). Similar to the results for SW1783, it was confirmed that PC also acts against *V. cholerae* infection.

### 2.3. DN Analysis

#### 2.3.1. Alzheimer’s Diffusion Network (ADN) Construction

DN construction began with 14 proteins (layer 1) from the MDA results and 22 proteins (layer 5) from DEGs. Network construction identified 84, 169, and 24 proteins in layers 2, 3, and 4, respectively (Figure 3A). Finally, only nine proteins remained in layer 1, and eight remained in layer 5.

#### 2.3.2. Submodule Construction for Identifying MOA

Proteins belonging to the the KEGG Alzheimer’s pathway gene set (KAGS) were color-mapped to the ADN (Figure 3B). In layer 1, all nine genes were included in the gene set (9/9). The deeper the layer, the lower the probability of belonging to a gene set. Only 23 proteins (23/84) in layer 2, 23 proteins (23/169) in layer 3, and 1 protein (1/24) in layer 4 belonged to the gene set. In the last DEG layer, the gene set did not contain any proteins belonging to the KAGS (0/8).

Only proteins belonging to the KAGS were selected separately, and the network was reconstructed to confirm which submodules in layer 3 were associated (Figure 4B and Appendix A). As a result of this analysis, seven proteins were selected from layer 1 except for two without connections. In layer 2, the groups interacting with layer 1 were rearranged into MTOR, PIK3CD, IDE, NOS2, and CACNA1C groups. Layer 3 was also rearranged into MTOR, MTOR, PIK3CD, PIK3CD, MTOR, NOS2, and NOS2 groups based on major connectivity. Each group was linked to related AD submodules. The MTOR group was related to the Wnt signaling pathway and Zn to anterior axonal transport and autophagy impairment mechanisms (Appendix A). The MTOR and PIK3CD groups acted on the insulin signaling pathway (Appendix A), whereas the PIK3CD group acted on the AGE-RAGE signaling pathway (Appendix A). The MTOR and NOS2 groups were confirmed to be associated with apoptosis (Appendix A).

### 2.4. Antioxidant Activity of PC in Lipopolysaccharide (LPS)-Stimulated BV2 Microglial Cells

It is known that the intracellular Reactive Oxygen Species (ROS) is increased in BV2 microglial cells by LPS treatment [19]. In the present study, therefore, the antioxidant activities of PC in hot water (WEPC) and 70% ethanol (EEPC) were evaluated in LPS-stimulated BV2 cells. Intracellular ROS was measured in LPS-stimulated BV2 in the presence of 300 μg/mL WEPS or EEPS. NAC at 10 mM was used as a positive control with antioxidant activity. LPS treatment significantly increased the intracellular ROS in BV2 cells. However, pretreatment of PC or NAC could inhibit the LPS-mediated ROS production (Figure 4A). PC treatment alone did not change the intracellular ROS status. The activation of BV2 by LPS was confirmed by observing *Nos2* gene induction, which was significantly inhibited by PC or NAC treatment (Figure 4B, left). In addition, Real-Time Polymerase Chain Reaction (qPCR) analysis also showed that the mRNA of *Hmox1*, a phase II-detoxification enzyme, was increased by PC treatment in the LPS-stimulated BV2 as well as in the non-stimulated BV2 (Figure 4B, right). The activation of the Nrf2/Hmox1 antioxidant signaling pathway by PC was confirmed by Western blotting, showing that intracellular protein levels of Nrf2 and Hmox1 were increased by PC irrespective of the presence of LPS stimulation (Figure 4C).

## 3. Discussion

In this study, in silico MDA, in vitro DTA, and DN analyses were conducted to identify the various mechanisms of PC using PC terpenes. In clinical medicine, PC is effective against various conditions, including high blood pressure, bacterial infections, decreased metabolism, and mental disorders. In this study, the pharmacological MOA of PC was analyzed using various techniques to investigate its therapeutic effect in clinical medicine. PC was particularly effective in AD.

This study has four essential attributes. First, the therapeutic effects of lanostane-type terpenes and seco-lanostane-type terpenes, unique compounds of PC, were identified using MDA. Second, a new mathematical model-based analysis method was proposed to utilize even small amounts of compounds present in PC by analyzing dockable proteomes. Third, the clinical efficacy of PC and previous findings were verified using in silico MDA and in vitro DTA. Finally, to study the MOA of herbal medicines, the relationship between MDA was identified to predict the upstream MOA and DTA, and a DN-based analysis method for integrated analysis was proposed.

PC contains unique compounds, including lanostane and seco-lanostane terpenes. Terpenes are unsaturated hydrocarbons produced from natural products and exert antioxidant, anti-inflammatory, and anticancer effects [20]. Terpenes are present in other herbal medicines, and they have a scaffold structure different from lanostane-based terpenes, so they interact with different proteins. This approach is the basis for explaining the physiological activity of PC compared with that of other medicinal herbs. For example, triterpenes with five rings, such as oleanolic acid and hederagenin, can act on pathways involving cholinergic synapses, type 2 diabetes, and gonadotropin hormone-releasing hormone secretion [12]. In comparison, lanostane triterpenes most likely act on the Rap1 signaling pathway, nicotine addiction, neuroactive ligand–receptor, and cocaine addiction pathways (Figure 1A). The Rap1 signaling pathway is associated with the blood–brain barrier [21], and all other pathways are related to neuropsychiatric disorders. Interleukin-21, derived from the biological process of lanostane triterpenes, is well-associated with AD [22] (Figure 1B). Therefore, PC’s effectiveness against mental illnesses is its unique characteristic. As PC has a high possibility of pharmacological MOA, which animal and plant herbal medicines do not possess, research on PC is expected to become more important.

The present study confirmed the antioxidant potential of PC in the LPS-stimulated murine microglial cells (Figure 4). The Rap1 signaling pathway derived from docking analysis affects organ damage caused by oxidative stress [23,24] (Figure 1). ITGAL, a protein interacting with PC unique terpenes, is a key gene closely related to oxidative stress damage in periodontitis [25] (Table 1). GSEA for PC extracts confirmed many terms previously reported to be related to oxidative stress, including cancer, the MAPK signaling pathway, the Wnt signaling pathway, and the TGF-beta signaling pathway (Figure 2). In this study, the antioxidant effect of PC derived from in silico experiments and transcriptome analysis was confirmed through cell-based experiments. The computer-based mechanism prediction method shown in the study could lead to efficient antioxidant research.

Studies on multi-compound-based pharmacological mechanisms of herbs that do not consider the number of compounds face specific challenges. Owing to the characteristics of herbal medicines, analysis of the pharmacological MOA of herbs by considering the amounts of compounds can be challenging because the content of the compound may vary depending on the cultivation area and harvesting time. Several compounds present in PC were analyzed to overcome the limitations of quantitative analysis, and the results were interpreted based on the number of pathway appearances (Table 2). This approach assumes that an effect emerges when several compounds are combined. This is based on the fact that compounds with the same scaffold structure have similar pharmacological mechanisms [11,12] and that multiple-ligand-docking analysis is possible even with different structures [26]. However, because this approach considers each pathway to be acting on multiple compounds, it may be feasible for a broader range of pathways than those currently known. A permutation test was performed to overcome these drawbacks and limit the analysis of the MOA of the PC, which could be overanalyzed. The new mathematical model-based drug mechanism prediction method proposed in this study is significant because it indirectly estimates and uses compound amounts excluded in multi-component herbal medicine studies. From a clinical perspective, PC’s efficacy against high blood pressure, diabetes, and mental disorders has been predicted well. However, as the exact amount was not considered, MOA studies considering the amounts of herbal compounds should be conducted in future studies to reveal the MOAs of herbal medicine.

A characteristic of this study is that the upstream and downstream pharmacological MOAs in PC were significantly different. The MOAs for conditions, such as high blood pressure, diabetes, and psychiatric disorders were confirmed using MDA, while the MOAs for conditions such as psychiatric disorders, bacterial infections, and metabolism were identified using DTA. We analyzed the MOA of MDA and DTA differently; however, both analyses revealed effective pharmacological MOAs of PC as in previous studies on diabetes, immune responses, and lipid metabolism. PC exerts an anti-diabetic effect in diabetic mice through a mechanism involving glucose-lowering activity and sensitizing PPAR-gamma-independent insulin-mediated glucose uptake [27]. PC regulates immune responses by increasing natural killer cell activity and IFN-γ secretion and decreasing IL-4 and -5 secretion in mouse splenic T lymphocytes [28]. PC effectively reduces serum cholesterol and triglyceride levels in a high-fat-diet model and regulates lipid homeostasis and bile acid metabolism through the FXR/PPARα-SREBP signaling pathway [29]. Although the results from MDA and DTA differed, the pharmacological MOA for psychiatric disorders was consistent. PC is effective in treating psychiatric disorders. In studies to alleviate or modulate the symptoms of these disorders, such as apathy, depression, sleep disruption, and psychosis, which are the core features of patients with AD, Sun et al. identified that the PC extract could ameliorate cognitive function by reducing amyloid-β formation and improving amyloid-β clearance in an AD mouse model [30]. Another study showed that PC regulated the sleep architecture via a GABAA-ergic mechanism, which increased the protein expression of GAD65/67 and GABAA receptor subunits in the primary hypothalamic neuronal cells [31]. Overall, the pharmacological MOA derived from these two analyses can explain the herbal efficacy of PC and the latest research findings. Both MDA and DTA have advantages and disadvantages; however, as they are complementary, using both analyses together in future studies regarding MOA would prove helpful.

The need for a method that can integrate and explain the results of psychiatric disorders that are equally effective was predicted and confirmed in MDA and DTA, and a new analysis method based on a DN was introduced. A DN was constructed using the AD pathway, which confirmed that both MDA and DTA had significant effects, and the pathological MOA was successfully identified (Figure 4 and Appendix A). Two crucial findings were derived from this analysis. First, the AD pathway provided by KEGG comprises many biological terms, subnetworks, and specific subnetworks in which PC acts can be identified using DN. Second, terms related to the KEGG AD pathway gradually disappeared as the DN layer deepened; therefore, the disease-related KEGG pathway did not fully reflect the results of the transcriptome analysis (Figure 3). KEGG analysis outputs numerous biological terms, such as genes, proteins, RNAs, and compounds, and thus, is very complex and extensive. Hence, it can be speculated that including all the transcriptome analysis results would be difficult. The model proposed in this study can reduce and model complex physiological and pharmacological mechanisms and identify information not provided by existing databases; therefore, it can be used in drug research.

This study has some limitations. The polysaccharides in PC [32] could not be subjected to MDA because their molecular weights were too large. DTA assumes that polysaccharides with high molecular weights will be filtered out during extract sterilization; therefore, our research strategy using only terpenes is not problematic. Although the effective functions of polysaccharides are being studied [32], it is challenging to perform in silico analyses to predict the pharmacokinetic and pharmacodynamic properties of polysaccharides. Therefore, techniques to predict the pharmacological MOA of compounds with substantial molecular weights, such as polysaccharides, are needed in the future. Additionally, secondary metabolites of PC were not fully considered. Most secondary metabolism changes functional groups rather than the molecular backbone. Therefore, the existing research results can sufficiently encompass the effects of secondary metabolites. However, from an activity cliff perspective, effector mechanisms can have a significant impact. Future studies should consider secondary metabolites of PC.

In the present study, the therapeutic efficacy of PC was verified using in silico MDA and in vitro DTA assays. In particular, the MOA of PC, widely used in clinical medicine, was identified using various biological data. In summary, the pharmacological MOA of terpenes in PC was identified. In silico MDA was performed by collecting PC terpenes from the literature and herbal compound databases to identify extensive pharmacological MOAs of PC terpenes. In vitro DTA was performed to confirm the pharmacological MOAs of PC. The analysis showed that although the MDA and DTA results did not match, both determined the clinical efficacy of PC. DN was used to integrate the two different results. Thus, the upstream and downstream pharmacological MOA of PC terpenes were identified. This study used research methods that have not been used in previous studies. Based on the molecular scaffold-based quantitative structure–activity relationship theory, we present an analytical model to predict biological MOA by combining small amounts of compounds with other compounds. We present DN, a research model that can confirm the diffusion of drug action mechanisms by linking the upstream and downstream mechanisms of the pharmacological MOA of PC. The method used in this study is expected to help identify the pharmacological MOA of complex natural products.

## 4. Materials and Methods

### 4.1. Overview of the Study

In this study, we investigated the biological MOA of PC using the terpenes present in PC. First, compound information was collected from PubMed and herbal compound databases. To predict the upstream MOA of PC that could be explained by drug–target interactions, MDA was performed using human druggable proteomes and the collected terpenes. Next, to confirm the downstream MOA of PC that could be explained by transcriptome, DTA was performed using the HT29 and SW1783 cell lines. The HT29 and SW1783 cell lines were selected to confirm their potential in treating digestive disorders, and neuropsychiatric disorders, respectively. Finally, using the analyzed AD as the effective MOA point in both the MDA and DTA results, an AD-based DN linking the results from both analyses was established to identify the detailed pharmacological MOA of PC.

### 4.2. In Silico MDA

#### 4.2.1. Method for Collecting Data on Terpenes Present in PC

Various sources were used to collect as much PC terpene information as possible. First, the terpenes included in 23 papers found by searching for “*Poria cocos*” and “HPLC” in the PubMed database were collected. Terpenes not found in PubMed were added by referring to several PC compound review papers [13,14,15]. Finally, we added terpenes that are included in our database of herbal compounds, such as the Traditional Chinese Medicine Systems Pharmacology Database and Analysis Platform (TCMSP) [33], the Traditional Chinese Medicine Integrated Database (TCMID) [34], and the database of medicinal materials and chemical compounds in Northeast Asian traditional medicine (TM-MC) [35]. The collected PC terpenes were classified as lanostane-type triterpenes, seco-lanostane-type triterpenes, pentacyclic triterpenes, tricyclic diterpenes, sterols, and other classes using the Chemical Entities of Biological Interest database [36] and manual curation.

For the collected PC terpenes, InChiKey and 3D structural information were searched using the PubChemPy library and manually curated; 125 compounds whose 3D molecular structures were retrieved from the PubChem database [37] were selected as druggable compounds.

#### 4.2.2. Method of Collecting Data Regarding Human-Derived Druggable Proteomes

Druggable proteome information was collected from the Human Protein Atlas (HPA) [38]. Druggable compounds primarily act on human protein targets, such as enzymes, transporters, ion channels, and receptors. The proteome list consisted of 812 proteins categorized into enzymes, transporters, ion channels, and receptors known to interact with FDA-approved drugs provided by the HPA (Appendix A).

Structural information on human-derived proteins was obtained from the AlphaFold 2.0 (AF) database [39]. AF, a database that provides artificial intelligence-based protein structure prediction results, also provides structural prediction information for human-derived proteins. After downloading the human-derived protein information of AF, 869 proteins included in the HPA druggable proteome list were extracted and selected as druggable proteomes for molecular docking (Appendix A).

#### 4.2.3. Large-Scale Molecular Docking Method Using Terpenes and Druggable Proteomes

Druggable compounds and proteomes were transformed into the PDQBT format, a structure that can be used for MDA using OpenBabel software (v3.1.1) [40]. Large-scale MDA was calculated by considering all cases in which the selected compounds and proteins were paired (125 compounds × 869 proteomes = 108,625 pairs). MDA was performed using Python and AutoDock Vina software (v.4.2.6) [41]. The MDA parameters set were as follows: exhaustiveness, maximum value of 100; box size, maximum value (126, 126, 126).

In the compound–protein pair derived from the MDA results, 100 proteins with the lowest binding affinities for each of the 125 druggable compounds were selected as the main dockable proteomes for each compound. We calculated the probability of dockable proteomes interacting with each terpene class to identify the proteins that interact with each.

#### 4.2.4. ORA Using Unique Compounds of PC

ORA was performed using lanostane and seco-lanostane triterpenes, which are unique PC compounds. The 125 components subjected to the docking analysis were classified into lanostane, seco-lanostane, and non-lanostane groups. Interactions with terpenes were confirmed for each protein and the number of terpenes with which the proteins interacted in each terpene group was counted. The number of proteins counted was expressed as a probability by dividing the total number of proteins in each terpene group. Subsequently, proteins with a docking probability of 50% or more in the lanostane and seco-lanostane groups and those with a docking probability of 50% or less in the non-lanostane group were defined as proteins interacting with PC-specific terpenes. ORA was performed using the EnrichR platform for proteins interacting with PC-specific terpenes [42]. For ORA, KEGG [43], GO biological process [44], and OMIM disease [45] databases were used. The results with the top 10 combined scores were selected as the ORA results of the unique compounds of PC.

Docking results were visualized by selecting ITGAL with the highest docking probability from the lanostane group. One compound each was selected from the three groups. We selected eburicoic acid, poricoic acid A, and 7-Oxodehydroabietic acid as representative compounds of lanostane-type triterpene, seco-lanostane-type triterpene, and non-lanostane-type triterpene. The Discovery Studio Visualizer (v21.1.0.20) was used for visualizing 3D ribbons and 2D diagrams, and AutoDockTools (v1.5.6) was used for 3D molecular surfaces.

#### 4.2.5. ORA Using Main Dockable Proteomes

ORA was performed using the main dockable proteome of the 125 components. ORA used the EnrichR API [42] provided by the Python GSEApy library [46] and was used to perform the analysis using the “KEGG_2021_Human” gene set [43]. After ORA, pathways that met the criteria of the Benjamini–Hochberg (BH) procedure (*p* < 0.0001) for all 125 druggable compounds were considered statistically significant [47]. The frequency of all pathways found in the significant pathways of each component was summed, and the resulting value was defined as the degree of significant MOA of the PC terpenes.

However, this analysis could not eliminate selection bias arising from the druggable proteome list. Hence, relative statistical analysis was performed using a permutation test to remove selection bias [48]. A random gene set was created by randomly extracting 100 proteomes from the 869 druggable proteomes. A permutation list was constructed by grouping 125 random gene sets, and 1000 permutation lists were constructed. From the list of 1000 permutations, the same method used for docking-derived proteins was repeated to extract 1000 significant random pathways (125 random gene sets × 1000 permutations = 125,000 ORA scores). The number of occurrences of each significant pathway derived from docking was counted, and the same process was repeated for 1000 permutation sets. The sum of the significant pathways in the druggable proteomes was compared with that in the list of 1000 permutations. The rank of the sum of the significant pathways in the druggable proteomes among all the permutated significant pathways was the *p*-value. Fifty-four pathways for which the significant pathways appearing within the top 5% (*p* < 0.05) were selected as statistically significant final pathways.

### 4.3. Producing PC-Induced Transcriptomes in SW1783 and HT29 Cell Lines

#### 4.3.1. Chemicals and Reagents

Dulbecco’s modified Eagle medium (DMEM), phosphate-buffered saline (PBS), TrypLE Express, penicillin–streptomycin, and fetal bovine serum (FBS) were purchased from Gibco (Grand Island, NY, USA). Leibovitz’s L-15 medium was purchased from the American Type Culture Collection (ATCC, Manassas, VA, USA). The cell culture flasks and multiwell culture plates were purchased from Thermo Fisher Scientific (Waltham, MA, USA). Dimethyl sulfoxide (DMSO), wortmannin, LY294002, and thioridazine were purchased from Sigma-Aldrich (St. Louis, MO, USA), and QIAzol Lysis Reagent was purchased from Qiagen (Germantown, MD, USA). The EZ-Cytox cell viability assay kit was purchased from DoGen Bio (Seoul, Republic of Korea).

#### 4.3.2. Preparation of Hot Water and Ethanol Extracts of PC

Dried *P. cocos* Wolf (Polyporaceae) was supplied by Kwangmyungdang Medicinal Herbs Co. (Ulsan, Republic of Korea), and its morphology was carefully validated by Dr. Goya Choi from Herbal Medicine Resources Research Center, Korea Institute of Oriental Medicine (KIOM), Republic of Korea. A voucher specimen (2-22-0221) was deposited in the Korean Herbarium of the Standard Herbal Resources of KIOM (Naju, Republic of Korea). The crushed material was extracted at 100 ± 3 °C for 3 h using a water reflux system (MS-DM609, Misung Scientific, Yangu, Republic of Korea or macerated in 70% ethanol for 1 h under an ultrasonication system (VCP-20, Lab companion, Dajeon, Republic of Korea) twice and filtered through a 5 µm cartridge filter (KOC Biotech, Daejeon, Republic of Korea). The filtrates were concentrated at 40 °C using a rotary evaporator (Ev-1020t, SciLab, Seoul, Republic of Korea) and then freeze-dried (LP20, ilShinBiobase, Dongducheon, Republic of Korea). The yields of WEPC and EEPC were 2.54% and 4.91%, respectively, and the final extracts were homogenized and stored in an airtight container in a cold room at 4 °C. For the in vitro investigation, homogenous PC extract was vigorously vortexed for 30 min at room temperature in PBS (Thermo Fisher Scientific, Rockford, IL, USA) containing 2% DMSO and then sterilized by filtering through a 0.22 µm membrane. The stock solution of PC (10 mg/mL) was aliquoted in a 1.5 mL tube and stored at −80 °C until use.

#### 4.3.3. Cell Culture

Human astrocytoma cell line SW1783 (HTB-13) and human colorectal adenocarcinoma cell line HT29 (HTB-38) were purchased from ATCC. SW1783 cells were maintained in Leibovitz’s L-15 medium supplemented with 10% (*v*/*v*) heat-inactivated FBS, 100 IU/mL penicillin, and 100 mg/mL streptomycin at 37 °C without CO_2_. HT29 cells and BV2 cells were maintained in DMEM supplemented with 10% (*v*/*v*) heat-inactivated FBS, 100 IU/mL penicillin, and 100 mg/mL streptomycin at 37 °C in a 5% CO_2_ incubator. The cells were sub-cultured every 3–4 days, depending on the cell density.

#### 4.3.4. Drug Treatment

One day before drug treatment, SW1783 and HT29 cells were plated at 1.5 × 10^5^ and 5 × 10^5^ cells/well, respectively, in a 6-well plate containing a 3 mL growth medium. The cells were exposed to 20 (Low), 100 (Intermediate), and 500 (High) µg/mL of PC by treatment with 150 µL per well of 0.4, 2, and 10 mg/mL PC. There was no cytotoxicity at a high dose (500 µg/mL), which was confirmed using the EZ-Cytox cell viability assay kit. PBS with 2% DMSO was used as the vehicle, and wortmannin, LY294002, and thioridazine were treated at a concentration of 10 µM as a positive control. After 24 h of drug treatment, the cells were washed thrice with ice-cold PBS. The total cell lysate was prepared using QIAzol Lysis Reagent and stored at −70 °C until RNA extraction.

#### 4.3.5. RNA Preparation for RNA-Seq

SW1783 and HT29 cells were subjected to total RNA extraction using the QIAzol Lysis Reagent, according to the manufacturer’s instructions. The concentration of isolated RNA was determined using an Agilent RNA 6000 Nano Kit (Agilent Technologies, Waldbronn, Germany). RNA concentration was determined using a Quant-it RiboGreen RNA Assay Kit (R11490, Thermo Fisher Scientific), and RNA quality was assessed by determining the RNA integrity number (>7) and 28S:18S ribosomal RNA ratio (>1.0) using the 2100 Bioanalyzer Instrument (Agilent Technologies). Total RNA was processed to prepare an mRNA sequencing library using the MGIEasy RNA Directional Library Prep Set (#1000006386; MGI Tech, Shenzhen, China) according to the manufacturer’s instructions. The library was quantified using the QuantiFluor ssDNA System (E3190, Promega, Fitchburg, WI, USA). The prepared DNA nanoballs were sequenced on an MGISeq system (MGI Tech) with 100 bp paired-end reads.

#### 4.3.6. RNA-Seq Preprocessing and Analysis of Differentially Expressed Genes (DEGs)

The quality of raw RNA-seq data was evaluated using FastQC (v0.11.9). Adapter sequences were removed from the reads using Trim Galore (v0.6.6). The cleaned reads were aligned to the human reference genome GRCh38 (hg19) using STAR (v2.7.9.a) with default parameters. The transcript abundance per gene was quantified using RSEM (v1.3.3) with the gene annotation GRCh38.84. The expected read counts and transcripts per million were used as the gene expression levels for further analyses. Differential gene expression analysis between the treatment and vehicle groups was conducted using the Wald test implemented in the R package DESeq2 (v1.38.2). The Wald statistic was used to rank genes, and this ranked list of genes was employed for gene set enrichment analysis (GSEA). The aforementioned analysis of RNA-seq data was performed using R (v4.2.2, R Foundation for Statistical Computing, Vienna, Austria). The RNA sequence data were deposited in the Gene Expression Omnibus with accession numbers GSE232862 and GSE232868.

### 4.4. GSEA and Cluster Analysis of Pathways

GSEA was performed using the DTA results obtained from the cell lines [49]. GSEA was performed for curated KEGG pathway gene sets in the Molecular Signature Database (MSigDB v7.5.1) [50] using the fgsea package (v1.24) [51] in R (v4.2.2) with parameters of minimum size 15, maximum size 500, and 100,000 permutations. The statistical significance of the GSEA results was evaluated by adjusting the *p*-value using the BH procedure. GSEA results were visualized as a heatmap using Pheatmap (v.1.0.12), and only pathways with results satisfying at least one statistical significance in the three PC concentrations were included in the heatmap. A pathway heatmap was used for hierarchical clustering analysis using correlation distance and complete methods [52].

### 4.5. DN Construction

#### 4.5.1. Method for Construction of AD DN Based on the Disease Pathway

A DN was constructed to explain the connection between the MDA and DTA results. DN was created using KAGS, which is a significant pathway in both MDA ORA and DTA GSEA results. Starting from the MDA results, a new layer was created with proteins exceeding the score (*n* ≥ 950) based on the protein–protein interaction (PPI) combined score provided by the STRING database [53]. The endpoints of the network used DEGs derived from the DTA results.

For the MDA-based proteins used in the network, all proteins predicted to act on the AD pathway in the ORA analysis were collected from the 125 compounds used for docking. Subsequently, 14 proteins with an appearance frequency of 20% or more were selected as proteins based on MDA (*n* ≥ 25). The DEGs used in the network were found to have a significant effect on the AD pathway, and the results for the intermediate concentration were used. Twenty-two proteins with an expression value greater than log2(0.5) or less than −log2(0.5) were selected as DEGs with an adjusted *p* < 0.05 in the intermediate concentration. The protein set for the layer construction was constructed by downloading human protein information from the STRING database.

The network has five layers (Figure 5). The first and fifth layers comprise MDA-based proteins and DEGs, respectively. In the protein set, excluding the proteins included in the first and fifth layers, those with a PPI combined score of 950 or higher were selected as the proteins of the second layer. A network was built in the first and second layers by connecting interactions with a PPI score of 950 or higher. The third and fourth layers were constructed similarly to the second layer, but all proteins that appeared once in the network were excluded from layer construction. The basic shape of the DN was completed by connecting interactions with PPI scores of 950 or higher between proteins belonging to each protein set in the fourth and fifth layers (Step 1).

The network was then updated using the backpropagation method. When the fourth and fifth layers were connected, not all proteins belonging to the fourth layer were connected to the fifth layer. All the nodes in the fourth layer, except those with at least one connection with the fifth layer, were deleted. Similarly, the DN was completed by sequentially deleting the protein nodes of the third, second, and first layers (Step 2). Network construction was performed using Python (v3.10.12), and network visualization was performed using Cytoscape software (v3.9.1) [54].

#### 4.5.2. Submodule Construction Method for Identification of MOA

A new DN was reconstructed using KAGS to confirm the pathological MOA of AD in the constructed DN. First, proteins belonging to the KAGS were color-mapped onto the ADN to determine how many AD-related genes were included. Next, the network was reconstructed in the ADN using only the proteins included in KAGS. All nodes, except for proteins included in the KAGS, were removed from the ADN. Subsequently, the nodes with no edges were removed. The reconstructed network was relocated for each layer. The second layer was rearranged into five groups based on its connection to the same proteins that belonged to the first layer. The third layer was also divided into five groups in a similar manner. The third layer groups were used to identify the MOA points of PC in the KEGG pathways and KEGG Mapper. The group that acted on the submodule in the pathway was connected by adding submodule nodes. All the above visualizations were performed using Cytoscape software (version 3.9.1).

### 4.6. Determination of Antioxidant Activity of PC

#### 4.6.1. Quantification of Intracellular ROS

The intracellular ROS was measured in LPS (Sigma-Aldrich)-stimulated BV2 microglial cells using an ROS-sensitive fluorescent indicator, 2′,7′-dichlorofluorescein diacetate (DCF-DA, Thermo Fisher Scientific) as described in the manufacturer’s protocol. In brief, BV2 cells were plated at 5 × 10^4^ cells/well in a black, clear-bottomed 96-well plate. After a 4 h incubation, the cells were pretreated with 300 μg/mL PC or 10 mM N-acetyl-L-cystein (NAC, Sigma-Aldrich) for 1 h and then stimulated with 250 ng/mL LPS for another 2 h. Then, cells were washed once with Hanks’ balanced salt solution containing Ca^2+^ and Mg^2+^ (HBSS) and incubated in HBSS containing 10 μM DCF-DA at 37 °C for 30 min. After two-time washes with HBSS, the fluorescent images were captured at 100 × magnification under a fluorescent microscope (TH4-200, Olympus Optical Co., Tokyo, Japan). To quantify fluorescence intensity, the cells were lysed in 0.1 mL RIPA lysis buffer (Thermo Fisher Scientific) for 5 min with gentle shaking, and the fluorescence was measured at an excitation of 485 nm and emission of 535 nm using a Spectramax i3 microplate reader (Molecular Devices, Sunnyvale, CA, USA).

#### 4.6.2. Real-Time Polymerase Chain Reaction

The gene expression of *Nos2* and *Hmox1* in LPS-stimulated BV2 in the presence of PC was determined by qPCR. In brief, BV2 cells were plated at 5 × 10^4^ cells/well in a 6-well plate and incubated for 4 h. Then, the cells were pretreated with 300 μg/mL PC or 10 mM NAC for 1 h and stimulated with 250 ng/mL LPS for another 24 h. Total RNA was isolated from the cells using an RNA-SpinTM Total RNA Extraction Kit (iNtRON Biotechnology, Seongnam, Republic of Korea), and cDNA was synthesized with 1 μg of total RNA using a High-Capacity cDNA Reverse Transcription Kit (Thermo Fisher Scientific). qPCR was performed using a 2 × Power SYBR Green PCR Master Mix (Thermo Fisher Scientific) in a CFX96 Touch Real-Time PCR Detection System (Bio-Rad, Hercules, CA, USA). The primer sequences were for qPCR as follows: *Nos2*, forward GAATCTTGGAGCGAGTTGTGGA, reverse GTGAGGGCTTGGCTGAGTGAG; *Hmox1*, forward CCAGGCAGAGAATGCTGAGTTC, reverse AAGACTGGGCTCTCCTTGTTGC; *Gapdh*, forward AAGGTGGTGAAGCAGGCAT, reverse GGTCCAGGGTTTCTTACTCCT. The expression of the target genes was normalized by that of the *Gapdh* housekeeping gene.

#### 4.6.3. Western Blotting

The expression of the Nrf2 and Hmox1 protein was determined by Western blotting. In brief, BV2 cells were plated at 5 × 10^5^ cells/well in a 6-well plate and incubated for 4 h. Then, the cells were pretreated with 300 μg/mL PC or 10 mM NAC for 1 h and stimulated with 250 ng/mL LPS for another 24 h. The whole cell lysate was prepared in RIPA lysis buffer containing the Halt Protease and Phosphatase Inhibitors (Thermo Fisher Scientific). The protein concentration was determined by bicinchoninic acid assay (Thermo Fisher Scientific). Equal amounts (20 μg) of whole cell lysate were subjected to 4–15% gradient SDS-PAGE. The separated proteins were electroblotted on the nitrocellulose membrane (Bio-Rad). The protein-blotted membrane was blocked for 1 h with an EzBlock Chemi solution (ATTO Korea, Daejeon, Republic of Korea) and then incubated overnight at 4 °C with a primary antibody diluted in a blocking solution. After three-time washes with 0.1% Tris-buffered saline-Tween 20 (TBST), the membrane was incubated with horseradish peroxidase (HRP)-conjugated secondary antibody for 1 h. After three-time washes with TBST, the target bands were visualized using the chemiluminescent substrate solution (SuperSignal West Pico Plus, Thermo Fisher Scientific). The images of protein bands were taken using a Fusion SL (Vilber, Collégien, France) and their intensities were quantified using the ImageJ (v1.53k, NIH, Bethesda, MA, USA). The expression of target proteins was normalized with β-actin. The antibodies used for Western blotting were as follows: Nrf2 (#12721, Cell Signaling Technology, Beverly, MA, USA), Hmox1 (Ab13248, Abcam, Cambridge, UK), β-actin (A1978, Sigma), HRP-conjugated mouse IgGκ-binding protein (sc516102) and HRP-conjugated anti-rabbit IgG (sc2357) from Santa Cruz Biotechnology (Santa Cruz, CA, USA).

## Figures and Tables

**Figure 1 ijms-25-04636-f001:**
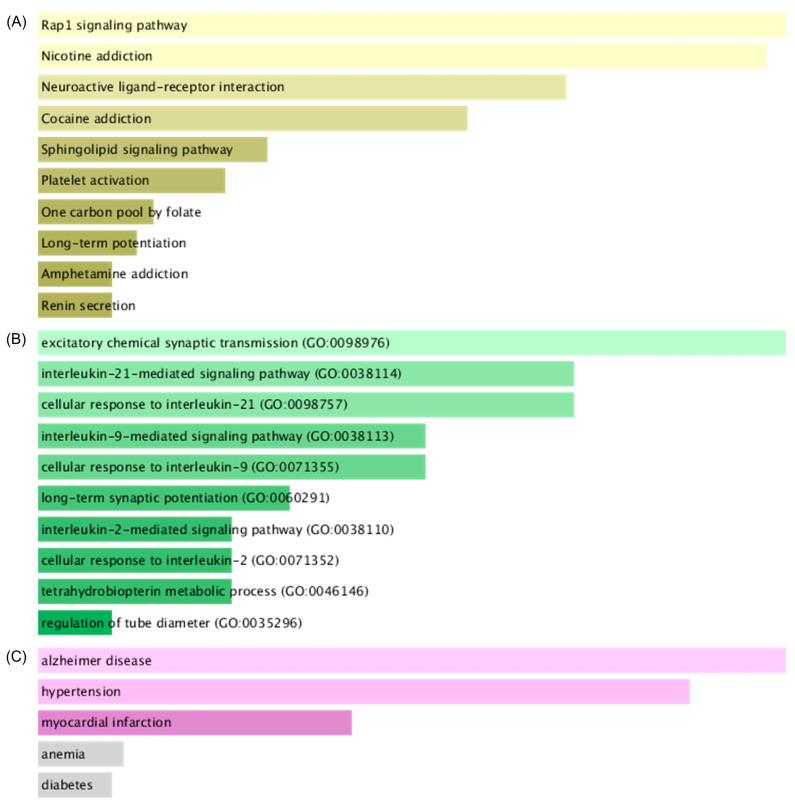
ORA results using lanostane terpenes and seco-lanostane terpenes. ORA was performed on the EnrichR platform using proteins that specifically bind to lanostane and seco-lanostane terpenes, which are unique terpenes of PC. The lightest color has a higher combined score, and as the color becomes darker, the combined score decreases. Gray are terms that did not pass the significance test. (**A**) ORA results using the KEGG gene set. (**B**) ORA results using the GO biological process gene set. (**C**) ORA results using the OMIM disease gene set. ORA, Over-representation analysis. PC, *Poria cocos* F.A. Wolf; KEGG, Kyoto Encyclopedia of Genes and Genomes; GO, Gene Ontology; OMIM, Online Mendelian Inheritance in Man.

**Figure 2 ijms-25-04636-f002:**
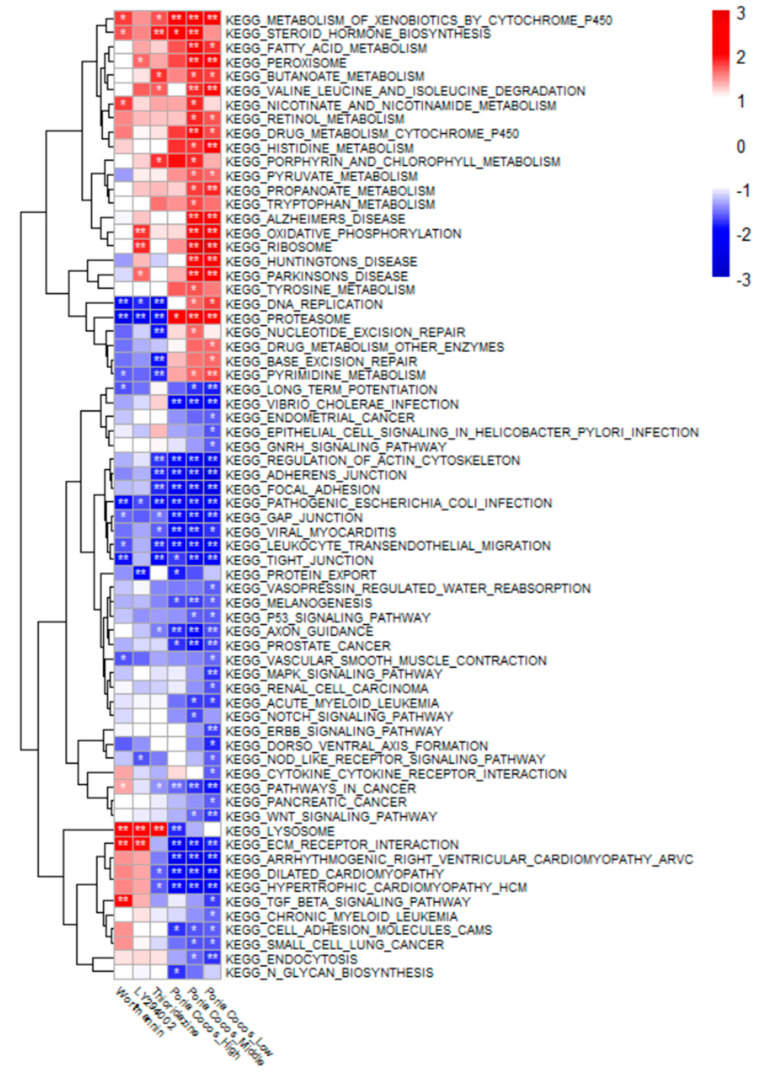
GSEA results using the SW1783 cell line–based PC-induced transcriptomes. GSEA was performed using the KEGG pathway gene set. Poria Cocos_High, Poria Cocos_Middle, and Poria Cocos_Low are the results of analyzing the gene expression values obtained by treating the PC water extract with high (500 µg/mL), medium (100 µg/mL), and low (20 µg/mL) concentrations, respectively. Wortmannin, LY294002, and thioridazine are positive controls. Wortmannin and LY294002 are known to inhibit inflammation by acting on the cell cycle, and thioridazine is a known dopaminergic receptor antagonist. GSEA, gene set enrichment analysis. PC, *Poria cocos* F.A. Wolf; KEGG, Kyoto Encyclopedia of Genes and Genomes. * *p* < 0.05, ** *p* < 0.01.

**Figure 3 ijms-25-04636-f003:**
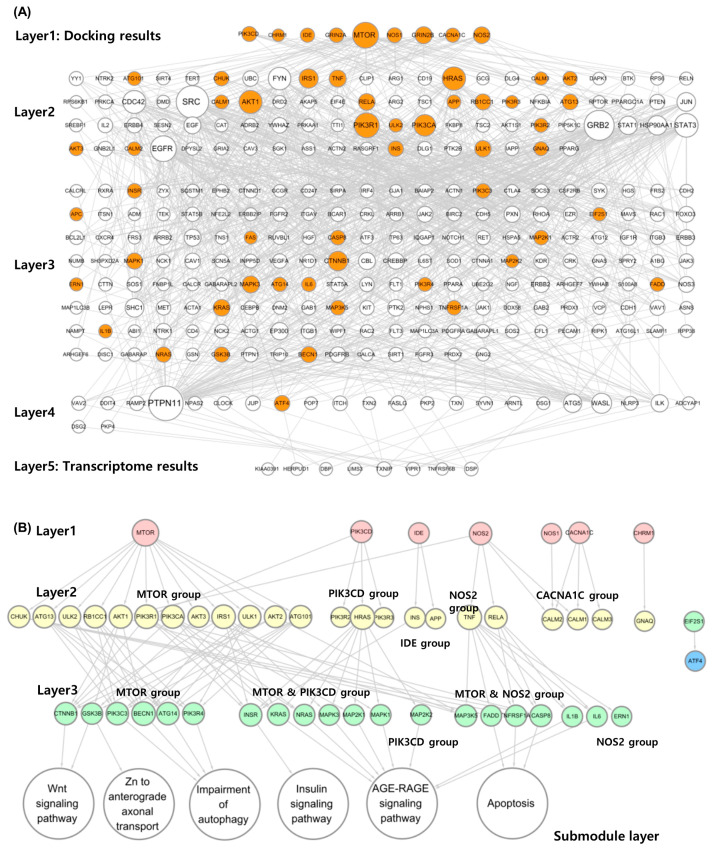
MOA analysis of PC using AD–related DN. (**A**) DN associated with AD. In layer 1, terms related to AD were selected from the results of MDA, and in layer 5, DEGs were selected and a DN was built using them. Orange circles indicate proteins included in the KEGG AD gene set. (**B**) MOA analysis result of PC using AD–related DN. Each layer was rebuilt using only the orange circles in the network above. Layers 2 and 3 were grouped so that it was possible to identify which protein signal transduction was initiated in layer 1. The AD submodule that each group specifically acts on was confirmed using KEGG Mapper. MOA, mode of action; PC, *Poria cocos* F.A. Wolf; AD, Alzheimer’s disease; DN, diffusion network; MDA, molecular docking analysis, DEGs, differentially expressed genes; KEGG, Kyoto Encyclopedia of Genes and Genomes.

**Figure 4 ijms-25-04636-f004:**
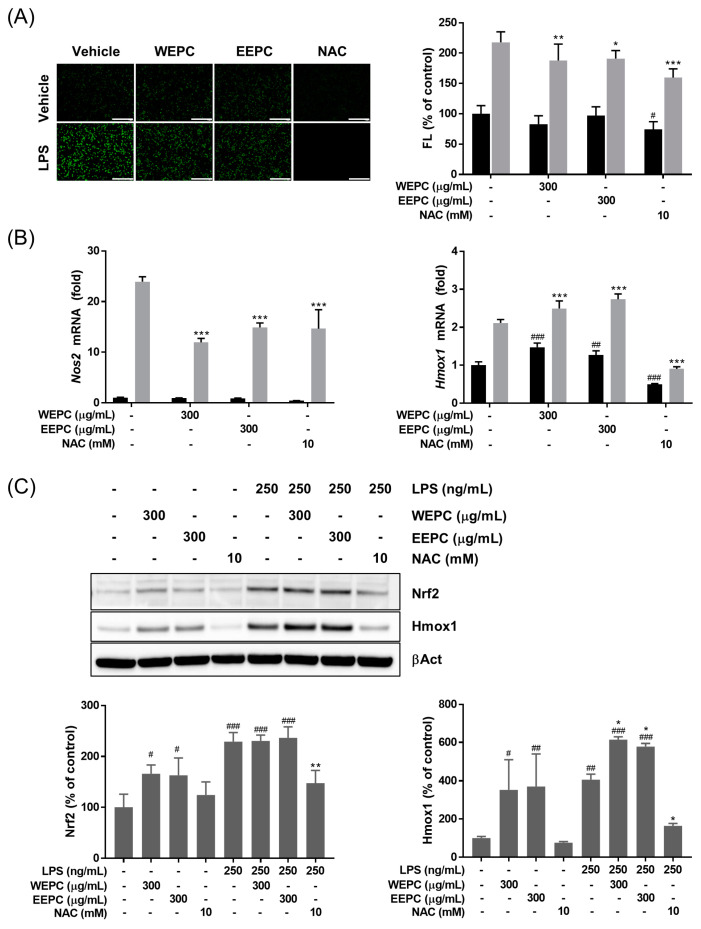
Antioxidant activity of PC. (**A**) Pretreatment of water (WEPC) and 70% ethanol (EEPC) extracts of PC inhibits LPS–mediated ROS production in the BV2 microglial cells. Intracellular ROS was monitored by DCF–DA staining (left, scale bar 400 μm) and their fluorescence intensities were quantified (right). Black bars, without LPS treatment; gray bars, with LPS treatment. (**B**) Changes in *Nos2* (left) and *Hmox1* (right) mRNA levels in BV2 microglial cells by PC in the absence (black bars) and the presence (gray bars) of LPS stimulation were determined by qPCR. (**C**) Activation of Nrf2/Hmox1 antioxidant signaling pathway by PC in the absence and the presence of LPS stimulation was observed by Western blotting (top). Expression of Nrf2 (left) and Hmox1 (right) proteins were quantified by image analysis. NAC was used as a positive control with antioxidant activity. β–actin was included as a loading control. Data are presented as means ± SD (*n* = 3 except for figure (**A**), *n* = 6). # *p* < 0.05, ## *p* < 0.01, ### *p* < 0.001 vs. vehicle-treated control without LPS treatment. * *p* < 0.05, ** *p* < 0.01, *** *p* < 0.001 vs. vehicle-treated control with LPS treatment.

**Figure 5 ijms-25-04636-f005:**
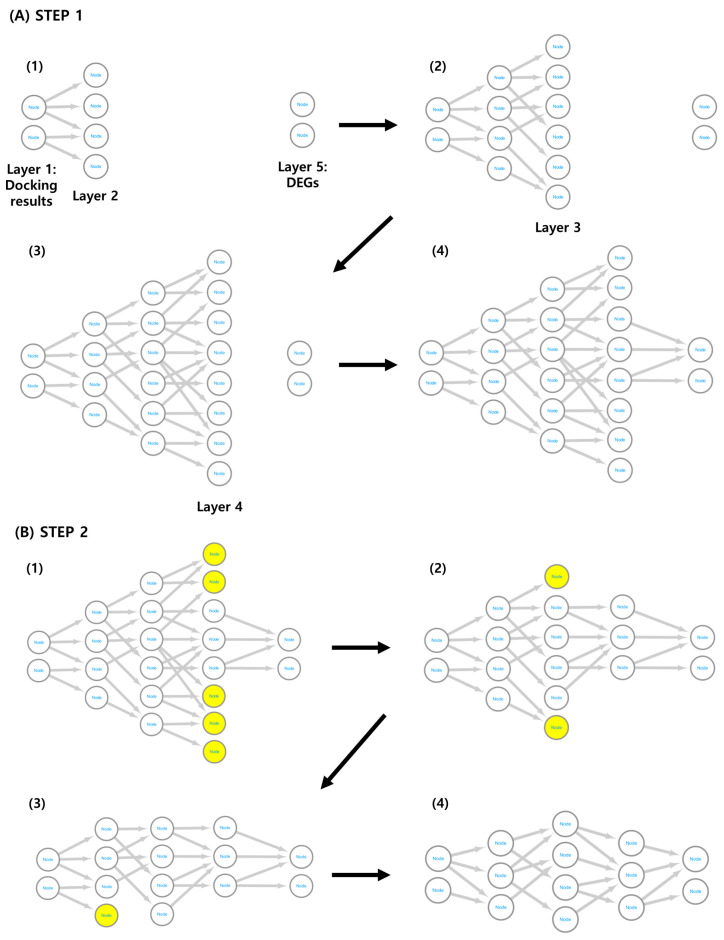
Method for constructing the DN. The DN was constructed to integrate results from MDA and DTA. (**A**) DN construction method. A new layer was assigned to the protein derived from the MDA results and the interacting proteins, and the process of connecting the previous layer and the new layer was repeated three times to add a total of three layers. In the fourth layer, the network was completed by adding connections that interact with genes derived from DTA. Numbers (1) through (4) indicate the order in which DNs are constructed. (**B**) DN modification method. In the fourth layer, nodes that did not have a connection with the last layer were deleted. In this manner, the network was completed by sequentially deleting nodes using the backpropagation method up to the second layer. Numbers (1) through (4) indicate the order in which DNs are constructed. Nodes marked in yellow indicate nodes that will be removed in the next sequence. DN, diffusion network; MDA, molecular docking analysis; DTA, drug-induced transcriptome analysis.

**Table 1 ijms-25-04636-t001:** Results of over-representation analysis using unique terpenes of *Poria cocos* F.A. Wolf.

Protein	Lanostane-Type Triterpenes	Seco-Lanostane-Type Triterpenes	Non-Lanostane-Type
ITGAL	0.910714	0.774194	0.263158
ITGAX	0.821429	0.806452	0.131579
GRIN2B	0.821429	0.903226	0.368421
CHRM4	0.803571	0.967742	0.394737
HCRTR1	0.785714	1	0.210526
SLC6A8	0.785714	0.903226	0.394737
SLC6A4	0.767857	0.709677	0.184211
GRIN2A	0.696429	0.967742	0.210526
HCRTR2	0.678571	0.967742	0.263158
PRKCZ	0.678571	0.741935	0.236842
PTGIS	0.678571	0.645161	0.184211
PRKCI	0.660714	0.83871	0.315789
DHFR	0.642857	1	0.473684
JAK3	0.642857	0.645161	0.394737
JAK1	0.642857	0.935484	0.394737
ACE	0.625	0.677419	0.315789
NOS3	0.607143	0.741935	0.447368
SLC22A6	0.607143	0.709677	0.289474
ADORA1	0.607143	0.935484	0.315789
TFRC	0.517857	0.870968	0.236842

**Table 2 ijms-25-04636-t002:** Top 25 prediction results for over-representation analysis using main dockable proteomes.

KEGG Pathway	Docking Count	*p*-Value
Calcium signaling pathway	125	0.006
cAMP signaling pathway	123	<0.001
Cholinergic synapse	115	<0.001
MAPK signaling pathway	115	<0.001
Serotonergic synapse	115	<0.001
Type II diabetes mellitus	114	<0.001
Hypertrophic cardiomyopathy	110	<0.001
Oxytocin signaling pathway	107	<0.001
GnRH secretion	102	<0.001
cGMP-PKG signaling pathway	101	<0.001
GABAergic synapse	100	<0.001
Adrenergic signaling in cardiomyocytes	99	<0.001
Circadian entrainment	98	<0.001
Arrhythmogenic right ventricular cardiomyopathy	92	<0.001
Dilated cardiomyopathy	91	<0.001
Cortisol synthesis and secretion	90	<0.001
Aldosterone synthesis and secretion	88	<0.001
Prion disease	87	<0.001
PI3K-Akt signaling pathway	84	<0.001
Pathways of neurodegeneration	80	<0.001
Dopaminergic synapse	79	<0.001
Renin secretion	75	<0.001
Chemical carcinogenesis	74	<0.001
Alzheimer’s disease	73	<0.001
Cushing’s syndrome	70	<0.001

## Data Availability

The raw sequence and processed data were deposited in the NCBI Gene Expression Omnibus (GEO, https://www.ncbi.nlm.nih.gov/geo/) with accession numbers GSE232862 and GSE232868 (accessed on 17 April 2024).

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
