# Peer review of "Effect of Poria cocos Terpenes: Verifying Modes of Action Using Molecular Docking, Drug-Induced Transcriptomes, and Diffusion Network Analyses"

_ijms, 2024, doi:10.3390/ijms25094636_

Round 1
Reviewer 1 Report
Comments and Suggestions for Authors
The authors performed an ensemble study about the effect of Poria cocos terpenes. The rationale of the work is understandable, but not always well explained and executed. The work is interesting but also confusing. The research design is not always appropriate and some results are missing. In my opinion, the work must be strengthened in the docking study, according to the major revisions proposed below.
Thus, the paper might be reconsidered after these major revisions.
Abstract:
The abstract is redundant and unclear. Rewrite it highlighting the results obtained.
Introduction:
Lines 59-60 “Several methods can be used to confirm the broad MOA of herbal medicines, such as molecular docking analysis (MDA) and drug-induced transcriptome analysis (DTA).” The docking predicts an affinity but does not confirm an activity.
Line 67 “MDA is capable of confirming direct drug-target interactions but not downstream drug actions.” As previously mentioned, docking predicted the binding pose.
Lines 73-74 “The upstream terpene mechanism was confirmed by drug-protein interactions through MDA”. Docking cannot confirm an activity.
Materials and Methods:
Lines 111-113 Provide the proteome list of 812 proteins in the supplementary.
Lines 114-115 The use of AlphaFold must be avoided if the experimental structure of the target is present.
Lines 117-119 It is not clear because proteins are 869.
Paragraph 2.2.3 The selection method is questionable in my opinion. A reference value should be defined for each biological target. Furthermore, the docking protocol must be validated.
Results:
It would be interesting to show some docking data such as scores and binding mode images.
Comments on the Quality of English LanguageExtensive editing of English language required.
Author Response
Reviewer 1.
The authors performed an ensemble study about the effect of Poria cocos terpenes. The rationale of the work is understandable, but not always well explained and executed. The work is interesting but also confusing. The research design is not always appropriate and some results are missing. In my opinion, the work must be strengthened in the docking study, according to the major revisions proposed below (in word file).
Thus, the paper might be reconsidered after these major revisions.
Response:
Thank you for your positive review. As mentioned, this is the first time that the efficacy of Poria cocos, which contains various compounds, has been verified using bioinformatics and statistical analysis methods, so it can be confusing. We used an almost first-of-its-kind method called large-scale molecular docking, which we believe may be new when interpreting data using the same methods used in statistical analysis or computer science. Nonetheless, we think you have pointed out exactly what we need.
We used a proofreading company to thoroughly proofread the English text, and the editing certificate is attached below.
Abstract:
The abstract is redundant and unclear. Rewrite it highlighting the results obtained.
Response:
We attempted to write the abstract again based on your suggestions. However, the 200 word-limit made it difficult to include all results. Nevertheless, we selected and modified the parts that we thought were important.
Revised abstract:
We characterized the therapeutic biological modes of action of several terpenes in Poria cocos F.A Wolf (PC) and proposed a broad therapeutic mode of action for PC. Molecular docking and drug-induced transcriptome analysis were performed to confirm the pharmacological mechanism of PC terpene, and a new analysis method, namely diffusion network analysis, was proposed to verify the mechanism of action against Alzheimer's disease. We confirmed that the compound that exists only in PC has a unique mechanism through statistical-based docking analysis. Also, docking and transcriptomic analysis results could reflect results in clinical practice when used complementarily. The detailed pharmacological mechanism of PC was confirmed by constructing and analyzing the Alzheimer's disease diffusion network, and the antioxidant activity based on microglial cells was verified. In this study, we used two bioinformatics approaches to reveal PC’s broad mode of action while also using diffusion networks to identify its detailed pharmacological mechanisms of action. The results of this study provide evidence that future pharmacological mechanism analysis should simultaneously consider complementary docking and transcriptomics and suggest diffusion network analysis, a new method to derive pharmacological mechanisms based on natural complex compounds.
Introduction:
Lines 59-60 “Several methods can be used to confirm the broad MOA of herbal medicines, such as molecular docking analysis (MDA) and drug-induced transcriptome analysis (DTA).” The docking predicts an affinity but does not confirm an activity.
Line 67 “MDA is capable of confirming direct drug-target interactions but not downstream drug actions.” As previously mentioned, docking predicted the binding pose.
Lines 73-74 “The upstream terpene mechanism was confirmed by drug-protein interactions through MDA”. Docking cannot confirm an activity.
Response:
We agree that predictions from docking analysis cannot fully confirm the mechanism. Therefore, when designing the study, we added diffusion network analysis to confirm the mechanism. In addition to the parts you pointed out, we have modified the parts that we believe had the same problem (Lines 83, 551, 625)
Materials and Methods:
Lines 111-113 Provide the proteome list of 812 proteins in the supplementary.
Lines 117-119 It is not clear because proteins are 869.
Response:
We will provide a list of 812 proteins (as of December 27, 2022) and a list of 869 modified AlphaFold proteins in the supplementary material. The number has increased from 812 to 869 because the AlphaFold database now provides 1,400 fragments for over 2,700 sequenced proteins (https://alphafold.ebi.ac.uk/faq). Because some druggable proteins are long, their fragment files have been expanded.
Lines 114-115 The use of AlphaFold must be avoided if the experimental structure of the target is present.
Response:
We agree that it may be better to use the experimental structures provided by PDB rather than AlphaFold. However, we used AlphaFold due to the following limitations:
- Not all proteins have experimental structures. The same pipeline should be used to perform statistical analysis on docking results. However, we used AlphaFold because performing the analysis using some experimental data and some predicted data may lead to different conclusions.
- The experimental structure file has a disadvantage that the zero point is not constant. When performing docking analysis with an experimental structure file, center_x, center_y, and center_z must be found and performed directly, which is difficult. There is no problem in performing an analysis using only a few proteins but it is difficult to determine the zero point of close to 1,000 proteins, and the act of determining the zero point requires arbitrary interpretation by the researcher, so it is not recommended.
- Additionally, the experimental structure file contains massive data that can only be confirmed under certain circumstances. For example, structural data used to observe interactions with a particular drug using X-ray crystallography includes the backbone of the drug. Additionally, proteins that act as transcriptional regulators sometimes exist in a modified state for interaction with nucleic acids. For statistical analysis, all proteins must be analyzed under the same circumstances but considering the above special situations, it is almost impossible to identify and preprocess all proteins one by one.
As the reviewer points out, because AlphaFold is a predicted protein structure, unexpected false positives may occur. So, to address this, we used a very high alpha level (p<0.0001) in our statistical analysis.
Paragraph 2.2.3 The selection method is questionable in my opinion. A reference value should be defined for each biological target. Furthermore, the docking protocol must be validated.
Response:
We agree with the reviewer. However, we believe that the analysis method we used is the best available. First, in docking analysis, it was difficult to define threshold values because there is no clearly agreed upon standard value. Also, the score varies greatly depending on the protein used for docking. For example, data from AlphaFold tends to have lower docking scores than those in PDB. And, the docking score depends on the size of the compound. For example, caffeic acid and gallic acid are good compounds for drug use but because these are small, they tend to have low docking scores. Therefore, not only is it difficult to apply specified reference values in batches, but accurate prediction results cannot be guaranteed. Therefore, we borrowed ideas from data science and used relative reference values for each compound. We attempted statistical-based verification of the docking protocol through the permutation method used for verification in data science (see Section 2.2.5).
Results:
It would be interesting to show some docking data such as scores and binding mode images.
Response:
We believe that because the validity of the data was verified based on statistical analysis, showing the analysis results using only a small number of proteins and components could lead to misinterpretation of the data. However, we agree that using images to display docking data makes it more interesting and accessible to readers. Accordingly, several images of docking analysis results were added to reflect the reviewer's opinion. Since statistical analysis using the docking analysis results is considered the most important data in docking analysis (Table 1), the results between the representative compounds of lanostane, seco-lanostane, and non-lanostane and ITGAL, at the top of Table 1, were visualized.
Revised method:
Docking results were visualized by selecting ITGAL with the highest docking probability from the lanostane group. One compound each was selected from the three groups. We selected eburicoic acid, poricoic acid A, and 7-Oxodehydroabietic acid as representative compounds of lanostane-type triterpene, seco-lanostane-type triterpene, and non-lanostane-type triterpene. Discovery Studio Visualizer (v21.1.0.20) was used for visualizing 3D ribbons and 2D diagrams, and AutoDockTools for 3D molecular surfaces
Revised result:
Visualization of docking analysis (Fig. S1) revealed the interaction of three compounds at the same position in ITGAL but the number of atoms interacting with the amino ac-id was different due to the difference in the size of the scaffold of compounds. There-fore, 7-Oxodehydroabietic acid, a non- lanostane compound, may preferentially act on proteins other than ITGAL despite functioning at the same location.

Reviewer 2 Report
Comments and Suggestions for Authors
The article under review presents an interesting and innovative approach tackling the bioactivity of a medicinal fungus, Poria cocos F.A.Wolf. Before publication, there are some issues to be dealt with.
Regarding the Introduction chapter:
- include at the beginning of the article and the abstract the complete current name of the fungus, Wolfiporia extensa (Peck) Ginns, and the complete synonym Poria cocos F.A.Wolf
- the fact that o Poria cocos is a fungus should be mentioned at the beginning of the introduction. The phrase in line 46 “A unique feature of PC is that it is a mushroom that belongs to the fungal family” should be deleted, as it is redundant. All mushrooms are fungi (and fungi are a Kingdom, not a family).
- the lines beginning with 37 : “For example, PC has a diuretic effect in Oryeong-san [4], an antidiarrheal effect in Samryeongbaekchul-san [5], digestive function in Yukgunja-tang [6], and a tranquilizing effect in Gwibi-tang [7]” should be replaced with: “PC is an active ingredient of the traditional Chinese remedies Oryeong-san exhibiting diuretic effect [4], Samryeongbaekchul-san having antidiarrheal effect [5], Yukgunja-tang that enhances digestive function [6], and Gwibi-tang with tranquilizing effect [7]”
- line 41: replace “Therefore, determining modes of action (MOA) of PC is essential to treat various diseases.” with : “Therefore, determining modes of action (MOA) of PC is essential to understanding the efficacy of the fungus in treating various diseases”
- start a new paragraph beginning with line 47 which contains a new idea – authors pass from the description of the fungal sclerotium to that of a pure compound; only pure compounds have a scaffold. “Previous studies have shown that drugs with similar scaffolds” should represent the beginning of a new paragraph
- line 55: replace “holistic” with “comprehensive”
- line 180: Preparation of hot water extract of PC – the section explains the obtainement of 2 extracts: one obtained by boiling in water, and another by ultrasonication in 70% ethanol – so the section title has to be modified to “Preparation of hot water and ethanol extracts of PC”
- explain the relevance of the author’s choice of these particular 2 cell lines: human astrocytoma cell line SW1783 (HTB-13) and human colorectal adenocarcinoma cell line HT29 (HTB-38), in the context of the research objectives
- if the effects of PC due to other secondary metabolites are not considered, that should be included in the discussion section
- in the conclusion section, there should be a clear take-home idea about the effects and modes of action that were revealed by the current research, as a paragraph or as a drawing. Currently, the last paragraphs (“In summary, …”) are rather a remembering of the research approach than a description of the main findings.
Comments on the Quality of English LanguageThere are some English language issues in the introduction section; the reviewer mentioned their correction.
Author Response
Reviewer 2.
The article under review presents an interesting and innovative approach tackling the bioactivity of a medicinal fungus, Poria cocos F.A.Wolf. Before publication, there are some issues to be dealt with.
Response:
Thank you for your positive review. We are confident that a better paper will be published, especially since the reviewers focused on editing the words and sentences used.
We used a proofreading company to thoroughly proofread the English text, and the editing certificate is attached below (in word file).
Regarding the Introduction chapter:
- include at the beginning of the article and the abstract the complete current name of the fungus, Wolfiporia extensa (Peck) Ginns, and the complete synonym Poria cocos F.A.Wolf
Response:
Thank you for your feedback. The abstract and main text have been revised and Poria cocos F.A. Wolf, the official name of Poria cocos, was used.
- the fact that o Poria cocos is a fungus should be mentioned at the beginning of the introduction. The phrase in line 46 “A unique feature of PC is that it is a mushroom that belongs to the fungal family” should be deleted, as it is redundant. All mushrooms are fungi (and fungi are a Kingdom, not a family).
Response:
Based on the reviewer's opinion, line 46 was removed, and instead the first sentence of the introduction was changed as “Poria cocos F.A. Wolf (PC), or Bokryeong, is a type of fungus that has long been used to treat a variety of diseases.”
- the lines beginning with 37 : “For example, PC has a diuretic effect in Oryeong-san [4], an antidiarrheal effect in Samryeongbaekchul-san [5], digestive function in Yukgunja-tang [6], and a tranquilizing effect in Gwibi-tang [7]” should be replaced with: “PC is an active ingredient of the traditional Chinese remedies Oryeong-san exhibiting diuretic effect [4], Samryeongbaekchul-san having antidiarrheal effect [5], Yukgunja-tang that enhances digestive function [6], and Gwibi-tang with tranquilizing effect [7]”
Response:
We tried to reflect the opinions of as many reviewers as possible. However, there seemed to be a bit of a problem, so we made a few changes. First, since PC is a medicine that contains one complex ingredient rather than a single compound, the content was changed to state that PC contains active compounds. Second, as the description doesn't match the previous sentence, the combination of P. cocos and other medicinal herbs was deleted from the previous sentence and the explanation was changed as follows:
“Because PC contains active compounds, it is used in traditional herbal remedies, such as Oryeong-san exhibiting diuretic effect [5], Samryeongbaekchul-san exerting anti-diarrheal effect [6], Yukgunja-tang that enhances digestive function [7], and Gwi-bi-tang with tranquilizing effect [8].”
- line 41: replace “Therefore, determining modes of action (MOA) of PC is essential to treat various diseases.” with : “Therefore, determining modes of action (MOA) of PC is essential to understanding the efficacy of the fungus in treating various diseases”
- start a new paragraph beginning with line 47 which contains a new idea – authors pass from the description of the fungal sclerotium to that of a pure compound; only pure compounds have a scaffold. “Previous studies have shown that drugs with similar scaffolds” should represent the beginning of a new paragraph
- line 55: replace “holistic” with “comprehensive”
Response:
The three comments above have been addressed in the revised manuscript based on your suggestions.
- line 180: Preparation of hot water extract of PC – the section explains the obtainement of 2 extracts: one obtained by boiling in water, and another by ultrasonication in 70% ethanol – so the section title has to be modified to “Preparation of hot water and ethanol extracts of PC”
Response:
It was revised to reflect the opinions of the reviewers. At the beginning of our study design, we wanted to conduct the experiments using water. However, there were limitations in using only a water extract, so ethanol was added. Although the content has been modified, the title has not been modified.
- explain the relevance of the author’s choice of these particular 2 cell lines: human astrocytoma cell line SW1783 (HTB-13) and human colorectal adenocarcinoma cell line HT29 (HTB-38), in the context of the research objectives
Response:
Reflecting the opinions of the reviewer, the following sentence was added to Section 2.1.1.
Revised method:
The HT29 and SW1783 cell lines were selected to confirm their potential in treating digestive disorders, and neuropsychiatric disorders, respectively.
- if the effects of PC due to other secondary metabolites are not considered, that should be included in the discussion section
Response:
In response to comments, the following has been added to the limitations under the discussion.
Revised discussion:
Additionally, secondary metabolites of PC were not fully considered. Most secondary metabolism changes functional groups rather than the molecular backbone. Therefore, the existing research results can sufficiently encompass the effects of secondary me-tabolites. However, from an activity cliff perspective, effector mechanisms can have a significant impact. Future studies should consider secondary metabolites of PC.
- in the conclusion section, there should be a clear take-home idea about the effects and modes of action that were revealed by the current research, as a paragraph or as a drawing. Currently, the last paragraphs (“In summary, …”) are rather a remembering of the research approach than a description of the main findings.
Response:
Thank you for your comment. Our study has slightly different features than existing studies on drug efficacy. Of course, in the case of new discoveries using P. cocos, the information you mentioned must be included. However, this study is closer to proposing a method to verify and reproduce existing pharmacological efficacy based on multiple active substances using bioinformatics methods. Therefore, our main conclusion is the research approach, and the reviewer's points can be considered to have conveyed our research points well.

Round 2
Reviewer 1 Report
Comments and Suggestions for Authors
The authors responded to my questions and I appreciated their explanation and the logic of the implementations made.
Reviewer 2 Report
Comments and Suggestions for Authors
The authors updated their manuscript according to the suggestions of the reviewer.